# Effects of specialist care lower limb orthoses on personal goal attainment and walking ability in adults with neuromuscular disorders

**Elza van Duijnhoven** [1,2] *, **Fieke S. Koopman** [1,2], **Hilde E. Ploeger** [1,2], **Frans Nollet** [1,2], **Merel-Anne Brehm** [1,2]

**1** Department of Rehabilitation Medicine, Amsterdam UMC location University of Amsterdam, Amsterdam, The Netherlands, **2** Amsterdam Movement Sciences, Rehabilitation & Development, Amsterdam, The Netherlands

* e.vanduijnhoven@amsterdamumc.nl

## Abstract

**Data Availability Statement:** All relevant data are within the paper and its Supporting Information files.

### Background

Lower limb orthoses intend to improve walking in adults with neuromuscular disorders (NMD). Yet, reported group effects of lower limb orthoses on treatment outcomes have generally been small and heterogeneous. We propose that guideline-based orthotic care within a multidisciplinary expert setting may improve treatment outcomes.

### Aim

To examine the effectiveness of specialist care orthoses compared to usual care orthoses on personal goal attainment and walking ability.

### Design

Cohort study.

### Population

Adults with NMD who experienced walking problems due to calf and/or quadriceps muscle weakness and were provided with a specialist care lower limb orthosis between October 2011 and January 2021.

### Methods

Three months after provision, the specialist care orthosis was compared to the usual care orthosis worn at baseline in terms of personal goal attainment (Goal Attainment Scaling (GAS)), comfortable walking speed (m/s), net energy cost (J/kg/m) (both assessed during a 6-minute walk test), perceived walking ability and satisfaction.

**Funding:** EvD was supported by ZonMw, The Netherlands Organisation for Health Research and Development.

**Competing interests:** The authors have declared that no competing interests exist.

## Results

Sixty-four adults with NMD were eligible for analysis. The specialist care orthoses comprised 19 dorsiflexion-restricting ankle-foot orthoses (AFOs), 22 stance-control knee-ankle-foot orthoses (KAFOs) and 23 locked KAFOs. Overall, 61% of subjects showed a clinically relevant improvement in GAS score. Perceived safety, stability, intensity, fear of falling and satisfaction while walking all improved ($p{\leq}0.002$), and subjects were satisfied with their specialist care orthosis and the services provided. Although no effects on walking speed or net energy cost were found in combined orthosis groups, specialist care AFOs significantly reduced net energy cost (by 9.5%) compared to usual care orthoses (from mean (SD) 3.81 (0.97) to 3.45 (0.80) J/kg/m, p = 0.004).

## Conclusion

Guideline-based orthotic care within a multidisciplinary expertise setting could improve treatment outcomes in adults with NMD compared to usual orthotic care by improvements in goal attainment and walking ability. A randomized controlled trial is now warranted to confirm these results.

## Introduction

Adults with neuromuscular disorders (NMD), including slowly-progressive neuromuscular diseases and peripheral nervous system injuries, often experience walking limitations [1] caused by a loss of lower extremity muscle strength [2]. Walking limitations typically include increased walking energy cost [3, 4], diminished walking speed [3, 5], pain [6], reduced balance and increased fall risk [7, 8], which together hinder daily physical activity [9] and negatively impact quality of life [10].

In NMD, lower limb orthoses for lower extremity muscle weakness are commonly applied [11], and intend to improve walking and walking safety [12–14]. The various types of available lower limb orthoses differ in terms of design, weight, material, and stiffness [15–19]. In people exhibiting distal leg muscle weakness, including weakness of the calf muscles, current practice includes provision of high orthopaedic shoes and ankle-foot orthoses (AFOs) (ISO 8549–1 [20]) that should ideally restrict excessive ankle dorsiflexion during the stance phase of gait [14, 18, 21]. Knee-ankle-foot orthoses (KAFOs) (ISO 8549–1 ]20]) are indicated in case of (additional) proximal leg muscle weakness, particularly of the quadriceps muscles, and aim to ensure weight-bearing stability during standing and walking [13, 22].

While AFOs and KAFOs reportedly improve walking energy cost [4, 18, 23], balance [24] and walking speed [25–27] in adults with NMD, available evidence is mainly derived from small, low-quality laboratory-based studies [28, 29]. Furthermore, effects across studies and individuals are inconsistent [25, 29], hampering translation to clinical decision making. In current practice, orthotic prescriptions in NMD are largely dependent on the local preference and experience of clinicians [21, 30]. Consequently, orthosis designs vary considerably [31] and patients show wide variation in responses [18], possibly due to an inadequate match between the orthosis design and the patient's individual underlying impairments [18, 32].

To facilitate individualized treatment guidance, a Dutch guideline for the prescription of lower limb orthoses in NMD was published in 2012 [33]. This guideline provides standardized protocols for each step of the care process according to the process description medical devices

[34]. It also contains decision schemes for the selection of an orthosis based on an objective measurement of the patient's gait deviations with a 3D gait analysis and of the underlying impairments in terms of extent and severity of muscle weakness and joint contractures. Following this guideline-based approach, the provision of custom-made AFOs and KAFOs by a multidisciplinary team in an orthosis expert centre (i.e. specialized orthotic care) reportedly improves gait biomechanics and reduces walking energy cost in polio survivors [23, 26]. However, little evidence is currently available concerning adults with diverse NMD, especially comparisons with usual care orthoses (UC orthoses), and the effects of specialist care orthoses on patient-reported outcomes such as personal goal attainment is also poorly defined [28]. To the best of our knowledge, only one NMD-related study has evaluated the effects of lower limb orthoses on personal goals, as measured by the goal attainment scale (GAS) [35], a patient-reported measure that captures the effects of orthotics in relation to individual patient priorities, as such providing important additional information [36].

In this study, we compared specialist care orthoses to UC orthoses (control condition at baseline) in terms of personal goal attainment, walking ability outcomes and satisfaction. The patient population consisted of adults with NMD experiencing walking problems due to lower extremity muscle weakness. We hypothesized that specialist care lower limb orthoses would improve treatment outcomes compared to UC orthoses.

## Materials and methods

### Study design

In this cohort study, we analysed data that were prospectively collected between October 2011 and March 2021 during orthotic care delivery in our university hospital outpatient polio and orthosis expertise rehabilitation clinic in Amsterdam, the Netherlands. All subjects gave informed consent for their data to be used for research purposes. The requirement for ethical review of the study under the Medical Research Involving Human Subjects Act in the Netherlands was waived by the medical research ethics committee of the Amsterdam UMC (University of Amsterdam) on October 21 2021. Reporting of the study was in accordance with the Strengthening the Reporting of Observational Studies in Epidemiology (STROBE) guidelines [37].

### Study population

On March 29 2021, we searched our orthotic database to identify adults with NMD who had been assessed in our gait lab for gait problems. Eligibility criteria were: (1) minimum age of 18 years; (2) weakness of the calf muscles (i.e. Medical Research Council (MRC) scale [38] score < 5 or not being able to make three single heel-rises on one leg) and/or quadriceps muscles (i.e. MRC score < 5); (3) provision of their first specialist care orthosis; and (4) currently wearing a UC orthosis. We excluded patients for whom no follow-up data were available concerning evaluation of their specialist care orthosis.

### Intervention

**Specialist care orthoses.**   Specialist care orthoses were prescribed by an experienced rehabilitation physician in accordance with the decision schemes of the Dutch national guideline for lower limb orthoses in NMD [33], which was implemented in our outpatient clinic in 2011 (see S1 Fig for a schematic presentation of the process description medical devices [34], on which the steps of the specialized care process according to the guideline were based). All orthoses, including AFOs, stance-control KAFOs (SC-KAFOs) and locked KAFOs were

custom-made of carbon composite by an experienced orthotic technician (OIM Orthopedie, Noordwijkerhout, The Netherlands). AFOs were either designed as hinged AFOs with dorsiflexion-stop, as spring-hinged AFOs or as dorsal leaf spring AFOs. SC-KAFOs were equipped with a mechanically or electronically-controlled knee joint or with a posteriorly offset knee joint.

Subjects visited our outpatient clinic for an intake and physical examination, including an assessment of muscle atrophy, deformities and range of joint motion, manual strength testing, sensory impairments and functional tests including single-leg heel rises and squats. Thereafter, we performed a baseline measurement with the patients' UC orthosis, including a 3D gait analysis along a 12-meter walkway including two force plates (OR6-7, AMTI, Watertown, MA, USA) recorded with a Vicon Vantage V5 motion capture system (VICON, Oxford, UK) (also including a barefoot condition), a 6-minute walk test (6MWT) to assess comfortable walking speed and net energy cost, and we asked the patient to complete questionnaires on perceived walking ability and satisfaction. The 3D gait data was analysed and interpreted in terms of gait kinematics and kinetics to determine the indication for a new orthosis as well as its design.

Subsequent visits were planned for casting and personal goal setting (with GAS), fitting of a thermoplastic foil test orthosis, fitting of the orthosis, delivery of the final orthosis, gait training by an experienced physiotherapist and a check-up 2 weeks after delivery. If adjustments were necessary, additional visits were planned, and when foot and/or ankle deformities necessitated the prescription of custom-made footwear, visits to a shoe technician were planned. Three months after final delivery of the orthosis, the follow-up measurement was performed with the patient wearing their specialist care orthosis and the results were evaluated together with the patient. When needed, the orthosis was adjusted based on the follow-up measurement.

**Usual care orthoses.**   Specialist care orthoses were compared to the UC orthoses that subjects were using at baseline. These included orthopaedic shoes or any type of prefabricated or custom-made AFO or KAFO.

## Measurements and data analysis

All measurements were collected, pre-processed and entered into an Open Clinica database by trained gait analysts. Outcomes analysed in this study included leg muscle strength (baseline only), GAS scores, comfortable walking speed, net energy cost, perceived walking ability and satisfaction (specialist care orthosis only).

**Leg muscle strength.**   During the medical examination, which was performed by a rehabilitation physician, muscle strength of the left and right hip flexors, extensors, abductors and adductors, knee extensors and flexors, ankle dorsiflexors and plantar flexors was manually assessed and scored according to the MRC scale [38, 39]. We calculated a MRC-sum score for each leg (range 0–40) by adding the individual scores from the eight muscle groups that were assessed.

**GAS score.**   A trained physiotherapist assessed the GAS at baseline, determining and scaling one or two personal goals selected together with the patient and based on the patient's goals for the specialist care orthosis in daily living [40]. GAS is an individualized measurement tool that facilitates comparison of treatment goal attainment for a diversity of goals within and between individuals. Attainment of personal goals was quantified on a 6-point scale, with predefined outcome levels: −3 = worsened, −2 = unchanged/baseline, −1 = somewhat less than expected, 0 = expected outcome, +1 = somewhat more than expected and +2 = much more than expected. An improvement of at least two GAS score points was regarded as clinically relevant [41]. Raw GAS scores were used for statistical analyses. In cases where two goals were set for one subject, a mean score was calculated.

**Walking speed and net energy cost.** Walking speed and net energy cost were assessed with a 6MWT at self-selected comfortable speed on an indoor oval track. Throughout the test, oxygen uptake ($VO_2$) and respiratory exchange ratios (RER) were measured breath-by-breath using an accurate telemetric gas analysis device (K5, Cosmed, Rome, Italy) [42]. Participants were instructed not to eat or drink sugar-containing beverages within 2 hours of the measurement. Prior to the 6MWT, subjects rested on a chair for 8 minutes while $VO_2$ and RER were measured. The assessment of walking energy cost during a 6MWT at comfortable speed has previously been shown to be reliable in individuals with NMD [43].

Mean $VO_2$ and RER values were derived from a visually identified steady-state period of at least 60 seconds during the last 3 minutes of the rest test and 6MWT. Using these values, energy consumption (in J/kg/s) during rest and walking were calculated based on the following formula: $((4.940 \times RER) + 16.040) \times VO_2$, where $VO_2$ is ml/kg/s [44]. To calculate net energy consumption, resting energy consumption was then subtracted from energy consumption while walking. Net energy cost (in J/kg/m) was calculated by dividing net energy consumption by walking speed during the 6MWT (in m/s).

**Perceived walking ability.** Using an in-house questionnaire, perceived walking ability was evaluated in terms of safety, stability, intensity, safety while walking on irregular surfaces, ascending and descending stairs, fear of falling and overall satisfaction while walking in daily living. Each item was rated on an 11-point scale, ranging from 0 (worst possible score) to 10 (best possible score).

**Satisfaction with the orthosis and services provided.** Satisfaction with the specialist care orthosis and provided services was assessed using the Quebec User Evaluation Satisfaction with assistive Technology questionnaire, in a valid and reliable Dutch version (D-QUEST) [45]. Mean scores were calculated (range 1–5) for both domains, excluding questions answered as 'not applicable'.

## Statistical analysis

SPSS version 26 (IBM SPSS, Chicago, Illinois, USA) was used for all statistical analyses and significance levels were set at $p < 0.05$. Subjects' demographic (e.g. gender, age) and clinical characteristics (e.g. diagnosis, leg muscle strength) were presented using descriptive statistics. Differences between specialist care and UC orthoses were analysed with paired samples t-tests for continuous outcomes (walking speed and net energy cost). When data were not normally distributed and for ordinal data (GAS scores, perceived walking ability outcomes), Wilcoxon signed-rank tests were used. In addition, as working mechanisms of orthoses differ, effects in subgroups of specialist care orthosis types (AFO, SC-KAFO, locked KAFO) on walking speed, net energy cost and perceived walking ability were separately explored with paired samples t-tests or Wilcoxon signed-rank tests.

## Results

We identified 140 adults with NMD in the orthotic laboratory database who used a UC orthosis at baseline and were provided with a specialist care orthosis for walking problems due to calf muscle weakness and/or quadriceps weakness. Seventy-six subjects were excluded as they were lost to follow-up or follow-up measurements were not yet available with their specialist care orthosis (Fig 1), leaving 64 subjects (35 males) with a mean (SD) age of 61 (12) years. Subjects were predominantly diagnosed with poliomyelitis (94%).

An overview of the UC orthoses that were worn at baseline and the provided specialist care orthoses, is presented in Fig 2. The UC orthoses included 37 orthopaedic shoes, 8 AFOs (2 hinged AFOs with dorsiflexion stop, 1 spring-hinged AFO, 3 dorsal leaf spring AFOs, 1 foot-

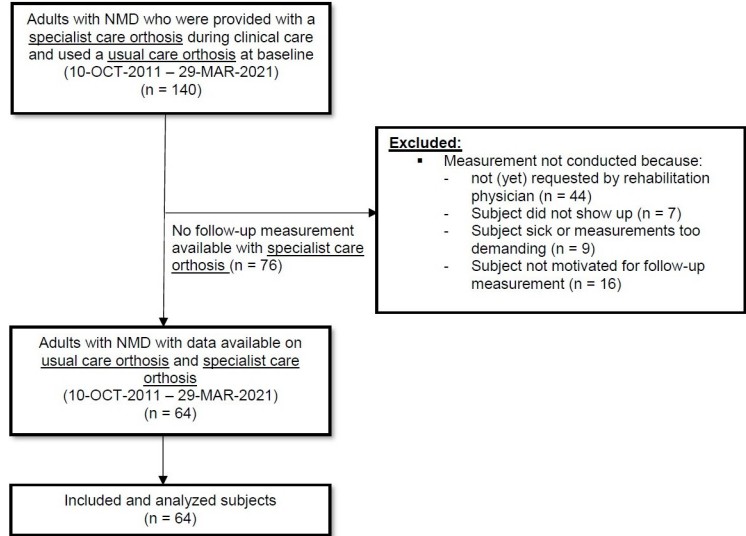

**Fig 1. Flow diagram of subject selection from the orthotic laboratory database, including reasons for exclusion.**

up), 6 SC-KAFOs (including one knee-orthosis) and 13 locked KAFOs, constructed from (a combination of) leather, metal, polypropylene, carbon composite or elastic materials. The UC orthoses were used for a median (IQR) 2.4 (5.4) years (data on nine subjects were missing).

Specialist care orthoses included 19 AFOs (12 hinged AFOs with dorsiflexion stop, 5 spring-hinged AFOs, 2 dorsal leaf spring AFOs), 22 SC-KAFOs and 23 locked KAFOs. In total, six subjects stopped using their specialist care orthosis before the follow-up measurement (1 spring-hinged AFO, 1 dorsal leaf spring AFO, 2 SC-KAFOs and 2 locked KAFOs). Median (IQR) time between baseline and follow-up measurements was 13.0 (8.0) months. Baseline demographics and clinical characteristics of all 64 subjects, and for subgroups according to specialist care orthosis type, are presented in Table 1.

| Specialist care orthoses → / Usual care orthoses ↓ | OS | AFO | SC-KAFO | Locked KAFO | Total |
|---|---|---|---|---|---|
| OS | - | 14 | 17 | 6 | 37 |
| AFO | - | **5** | 1 | 2 | 8 |
| SC-KAFO | - | - | **4\*** | 2 | 6 |
| Locked KAFO | - | - | - | **13** | 13 |
| Total | - | 19 | 22 | 23 | 64 |

**Fig 2. Overview of lower limb orthoses worn at baseline (UC orthoses) and at follow-up (specialist care orthoses).** Abbreviations; UC-orthosis: usual care orthosis, OS: orthopaedic shoes, AFO: ankle-foot orthosis, SC-KAFO: stance-control knee-ankle-foot orthosis, locked KAFO: knee-ankle-foot orthosis with locked knee joint. *One subject used a knee-orthosis at baseline.

**Table 1. Baseline demographics and clinical characteristics for all orthosis groups combined and for subgroups according to type of specialist care orthosis provided.**

| | All subjects n = 64 | AFO n = 19 | SC-KAFO n = 22 | Locked KAFO n = 23 |
|---|---|---|---|---|
| Gender (male / female) | 35 / 29 | 9 / 10 | 12 / 10 | 14 / 9 |
| Age in years *(mean (SD))* | 61 (12) | 62 (12) | 63 (12) | 58 (12) |
| Diagnosis (n) | Polio: 61 | Polio: 16 | Polio: 22 | Polio: 23 |
| | CMT: 3 | CMT: 3 | | |
| Use of assistive device* at baseline (no / yes) | 15 (23%) / 49 (77%) | 10 (53%) / 9 (47%) | 19 (86%) / 3 (14%) | 20 (87%) / 3 (13%) |
| Use of assistive device* at follow-up (no / yes) | 12 (19%) / 52 (81%) | 12 (63%) / 7 (37%) | 19 (86%) / 3 (14%) | 21 (91%) / 2 (9%) |
| Functional ambulation** level (at baseline)*** | In/around house: 29 (45%) | In/around house: 8 | In/around house: 10 | In/around house: 11 |
| | Rarely >1 km: 25 (38%) | Rarely >1 km: 6 | Rarely >1 km: 11 | Rarely >1 km: 8 |
| | Regularly >1 km: 5 (8%) | Regularly >1 km: 3 | Regularly >1 km: 1 | Regularly >1 km: 1 |
| Functional ambulation** level (at follow-up)*** | In/around house: 22 (34%) | In/around house: 3 | In/around house: 9 | In/around house: 10 |
| | Rarely >1 km: 25 (38%) | Rarely >1 km: 10 | Rarely >1 km: 8 | Rarely >1 km: 7 |
| | Regularly >1 km: 12 (19%) | Regularly >1 km: 5 | Regularly >1 km: 5 | Regularly >1 km: 2 |
| Unilateral / bilateral affected | 28 / 36 | 7 / 12 | 10 / 12 | 11 / 12 |
| MRC Plantar flexors most affected / least affected leg**** *(median [IQR])* | 0 [0–3] / 5 [4–5] | 0 [0–4] / 5 [4–5] | 0 [0–2.5] / 5 [4–5] | 0 [0–0] / 5 [5–5] |
| MRC Knee extensors most affected / least affected leg *(median [IQR])* | 2.5 [0.25–4] / 5 [4–5] | 4 [4–5] / 5 [5–5] | 3 [1.75–3.25] / 5 [5–5] | 0 [0–2] / 5 [4–5] |
| MRC sum-score most affected / least affected leg***** *(median [IQR]* | 16 [7–25] / 39 [34–40] | 27 [22–33] / 38 [35–40] | 17 [13.5–25.5] / 39 [34–40] | 8 [5–16] / 39 [32–40] |
| Frequency of wearing <u>specialist</u> care orthosis inside (n)*** | Always: 19 | Always: 3 | Always: 4 | Always: 12 |
| | Mostly: 26 | Mostly: 9 | Mostly: 14 | Mostly: 3 |
| | Rarely: 10 | Rarely: 6 | Rarely: 3 | Rarely: 1 |
| | Never: 4 | Never: 1 | Never: 1 | Never: 2 |
| Frequency of wearing <u>specialist</u> care orthosis outside (n)*** | Always: 34 | Always: 10 | Always: 9 | Always: 15 |
| | Mostly: 17 | Mostly: 7 | Mostly: 9 | Mostly: 1 |
| | Rarely: 5 | Rarely: 1 | Rarely: 3 | Rarely: 1 |
| | Never: 3 | Never: 1 | Never: 1 | Never: 1 |

Abbreviations; AFO: ankle-foot orthosis; SC-KAFO: stance-control knee-ankle-foot orthosis; locked KAFO: knee-ankle-foot orthosis with locked knee joint; CMT: Charcot-Marie-Tooth disease; MRC: Medical research council.

*Assistive devices used included 1 or 2 canes, 1 or 2 crutches or a walker.

**Functional ambulation level was evaluated at baseline and follow-up according to Brehm et al., 2020 [46].

***Based on 59 subjects as questionnaires of 5 subjects with a specialist care KAFO were missing.

****Based on 63 subjects as the MRC score of the plantar flexors was not assessed in 1 subject (stance-control KAFO user).

***** Based on 59 subjects as in 5 subjects not all muscle groups were tested (4 AFO users and 1 stance-control KAFO user).

## GAS score

GAS scores were available for 41 subjects and are reported in Table 2. GAS goals were not set in 11 subjects since the method had not yet been implemented at that point and in 8 subjects for unknown reasons. In a further four subjects, GAS goals were set at baseline but evaluation did not take place at follow-up for unknown reasons. Overall, the median [IQR] GAS score was significantly higher for specialist care compared to UC orthoses (from -2.0 [−2−−2] to -0.5 [-1.25–0.0]), $p<0.001$. The GAS score improved in 42 subjects (78%), was unchanged in 8 subjects (20%) and deteriorated in 1 subject. An improvement of at least 2 GAS points for at

**Table 2. GAS scores for specialist care orthoses at the follow-up measurement.**

| GAS score | Attainment level | Goal 1 (N) | Goal 2 (N) |
|---|---|---|---|
| -3 | Worsened | 1 | 1 |
| -2 | Unchanged (= starting point) | 11 | 8 |
| -1 | Less than goal | 10 | 7 |
| 0 | Goal | 7 | 9 |
| +1 | More than goal | 6 | 3 |
| +2 | Much more than goal | 6 | 4 |
| | **Total (N)** | 41 | 32 |
| | Median [IQR] | -0.5 [-2.0–1.0] | -0.5 [-2.0–0.0] |

Abbreviations; GAS: Goal attainment scaling, N: frequency of subjects, IQR: interquartile range. Note that only GAS scores at follow-up are presented, GAS scores at baseline were all set at -2 (starting point).

least 1 set goal is considered a clinically relevant change [41] and was achieved in 25 of 41 (61%) subjects, consisting of 3 of 8 (38%) AFO users, 13 of 19 (68%) SC-KAFO users and 9 of 14 (64%) locked KAFO users, respectively. When analysing all set goals (n = 73) classified by topic, clinically relevant improvements were achieved for 13 of 25 (52%) fatigue goals, 11 of 26 (42%) walking quality goals, 8 of 15 (53%) pain goals, 2 of 5 (40%) standing goals and 1 of 2 (50%) falling goals.

## Walking speed and net energy cost

Five subjects could not perform the 6MWT due to poor walking ability, and the 6MWT was not performed (either the baseline or follow-up measurement) in eight subjects due to technical problems with the Cosmed device. In the remaining 51 subjects, no significant differences were found in walking speed (from mean (SD) 0.84 (0.23) to 0.82 (0.22) m/s, *p* = 0.26, Fig 3A)

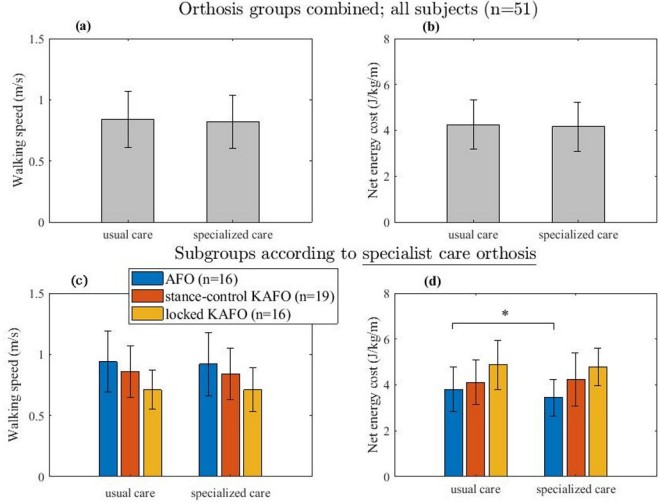

**Fig 3.** Bar plots of walking speed (m/s) and net energy cost (in J/kg/m) for the usual care orthoses and specialist care orthoses, respectively for all orthosis groups combined (upper panels A and B) and for subgroups of subjects provided with specialist care ankle-foot orthoses (AFOs), stance-control knee-ankle-foot orthoses (SC-KAFOs) and locked knee-ankle-foot orthoses (KAFOs) (lower panels C and D). Note that orthoses were categorized based on the provided specialist care orthoses, thus the usual care orthoses in panels C and D do not necessarily represent the same orthosis types. In all plots, error bars represent the standard deviation.

or in net energy cost (from 4.25 (1.08) to 4.16 (1.08) J/kg/m, $p$ = 0.296, Fig 3B) for specialist care versus UC orthoses.

While subgroup analyses based on type of specialist care orthosis found no significant differences in walking speed between the usual care AFOs and specialist care AFOs (from 0.94 (0.25) to 0.92 (0.26) m/s, $p$ = 0.442, Fig 3C), we did see a significant reduction in net energy cost, which reduced by 0.36 (0.42) J/kg/m (-9.5%) (from 3.81 (0.97) to 3.45 (0.80) J/kg/m, $p$ = 0.004, Fig 3D). Comparisons of other subgroups with specialist care SC-KAFOs or locked KAFOs revealed no significant differences in walking speed and net energy cost ($p$>0.397); see Fig 3C and 3D, respectively.

### Perceived walking ability

Data were missing on six individuals due to language barriers or unknown reasons. In the remaining 58 subjects, significant improvements were found for specialist care orthoses compared to UC orthoses in perceived safety (from (median [IQR] score) 5.0 [4.5–7.0] to 7.0 [5.5–8.0], $p$ = 0.001), stability (from 5.0 [3.5–6.0] to 7.0 [5.5–8], $p$<0.001), walking intensity (from 4.0 [3.0–5.0] to 6.0 [4.0–7.5], $p$ = 0.002), safety while walking on irregular surfaces (from 2.0 [1.0–3.5] to 4.0 [2.0–6.0], $p$<0.001), fear of falling (from 4.0 [2.0–7.0] to 6.0 [3.0–8.0], $p$ = 0.002) and overall satisfaction while walking in daily living (from 4.0 [3.0–5.0] to 6.0 [5.0–7.0], $p$<0.001). Scores for ascending and descending stairs did not differ between the UC and specialist care orthoses (from 4.0 [2.0–6.0] to 4.0 [2.0–7.0], $p$ = 0.354). Subgroup analyses according to type of provided orthosis showed that perceived stability and overall satisfaction while walking in daily living improved significantly in all specialist care orthoses groups compared to UC orthoses (all $p$≤0.031; Table 3). Furthermore, scores for ascending and descending stairs improved in the specialist care SC-KAFO group (from 3.0 [1.8–5.0] to 4.5 [1.8–7.0], $p$ = 0.042).

### Satisfaction with the specialist care orthosis and services provided

Data on the D-QUEST was available in 59 subjects, as the questionnaire was not completed by five subjects due to language barriers. The mean (SD) scores of specialist care orthoses were 3.7 (0.5) for orthosis-related aspects and 4.0 (0.6) for service-related aspects. As regards orthosis-related aspects, subjects were most satisfied with durability (80%) and safety (76%), and less satisfied with comfort (56%) and adjustability (59%). With respect to services provided, subjects were most satisfied with the professionalism (93%), follow-up (88%) and service in general (88%), but less satisfied with repairs (66%) and delivery (70%).

### Discussion

Our study showed that guideline-based orthotic care within a multidisciplinary expertise setting leads to the attainment of personal goals by a majority of adults with NMD who use a lower limb orthosis for walking problems. Perceived walking ability also improved compared to usual orthotic care. While no overall effects on walking speed or net walking energy cost were found, the specialist care AFO group showed a significant and clinically relevant reduction in net energy cost.

As regards to personal goal attainment (GAS), 78% of subjects improved overall and 61% of subjects improved by 2 or more GAS points on at least one goal, which is clinically relevant improvement. These findings are in line with the only other orthotic study to report GAS in NMD [35]. In our study, subjects provided with SC-KAFOs and locked-KAFOs in particular showed clinically relevant changes in their individual set goals, such as reducing fatigue and pain during walking, improving walking quality and preventing falls. This shows that,

**Table 3. Results for perceived walking ability for UC orthoses and specialist care orthoses in subgroups of subjects provided with specialist care AFOs, SC-KAFOs and locked KAFOs.**

| Perceived walking ability | Subgroup | Usual care orthoses | Specialist care orthoses | p-value |
|---|---|---|---|---|
| Safety | AFO (n = 17) | 5.0 [4.0–7.0] | 7.0 [5.0–8.0] | 0.084 |
| (0–10) | SC-KAFO (n = 22) | 5.0 [4.5–6.5] | 7.0 [5.75–8.0] | **0.025***  |
| | Locked KAFO (n = 19) | 6.0 [5.0–7.0] | 8.0 [7.0–9.0] | 0.067 |
| Stability | AFO (n = 17) | 5.0 [3.5–6.0] | 7.0 [5.0–8.0] | **0.014*** |
| (0–10) | SC-KAFO (n = 21) | 5.0 [3.0–6.0] | 7.0 [5.5–8.0] | **0.003*** |
| | Locked KAFO (n = 19) | 5.0 [4.0–6.0] | 8.0 [6.0–8.0] | **0.012*** |
| Walking intensity | AFO (n = 17) | 4.0 [2.5.-5.5] | 5.0 [4.0–7.0] | 0.056 |
| (0–10) | SC-KAFO (n = 22) | 4.0 [2.8–5.25] | 6.0 [2.8–8.0] | 0.088 |
| | Locked KAFO (n = 18) | 5.0 [3.0–5.25] | 6.5 [3.75–8.0] | 0.057 |
| Safety while walking | AFO (n = 17) | 3.0 [1.0–4.0] | 4.0 [2.0–5.0] | 0.121 |
| irregular surfaces (0–10) | SC-KAFO (n = 22) | 2.0 [1.0–2.25] | 4.0 [1.8–7.0] | **0.002*** |
| | Locked KAFO (n = 19) | 2.0 [1.0–4.0] | 4.0 [3.0–8.0] | 0.058 |
| Ascending and | AFO (n = 17) | 5.0 [3.0–6.5] | 3.0 [2.5–5.0] | 0.089 |
| descending stairs (0–10) | SC-KAFO (n = 22) | 3.0 [1.8–5.0] | 4.5 [1.8–7.0] | **0.042*** |
| | Locked KAFO (n = 19) | 4.0 [1.0–7.0] | 5.0 [2.0–8.0] | 0.218 |
| Fear of falling | AFO (n = 17) | 3.0 [1.0–7.5] | 6.0 [3.5–8.0] | **0.018*** |
| (0–10) | SC-KAFO (n = 22) | 4.0 [2.0–5.25] | 5.5 [3.0–8.0] | **0.004*** |
| | Locked KAFO (n = 19) | 6.0 [3.0–8.0] | 7.0 [3.0–9.0] | 0.490 |
| Overall satisfaction | AFO (n = 17) | 5.0 [3.0–5.5] | 6.0 [4.5–7.0] | **0.031*** |
| while walking (0–10) | SC-KAFO (n = 22) | 3.0 [2.75–5.0] | 7.0 [5.0–8.0] | **<0.001*** |
| | Locked KAFO (n = 19) | 4.0 [2.0–6.0] | 7.0 [5.0–8.0] | **0.004*** |

Scores range from 0 to 10, with 10 as most positive score and are presented as medians [IQR]. Abbreviations; AFO: ankle-foot orthosis, SC-KAFO: stance-control knee-ankle-foot orthosis, locked KAFO: knee-ankle-foot orthosis with locked knee joint.

*indicates significant difference between specialist care orthoses and usual care orthoses.

compared to available questionnaires, GAS is a meaningful approach when evaluating the effectiveness of orthotic treatment at the individual level, since questionnaires might fail to capture changes in personal goals resulting from orthotic treatment due to the heterogeneity of orthotic goals across individuals [47]. Overall, the clinically relevant improvements found for personal goals adds weight to the argument that specialist care orthoses improve functioning in daily living, especially regarding issues of primary importance to patients.

While a majority of subjects attained their personal goals, compared to UC orthoses, specialist care orthoses did not objectively improve walking speed or net energy cost, although subjects did report that walking required less effort. By contrast, an earlier study in polio survivors comparing carbon-composite specialist care orthoses to UC orthoses found a significant reduction (8%) in net energy cost [23]. However, in that study both the intervention and control conditions comprised locked KAFOs, whereas the specialist care orthoses provided in our study (including AFOs, SC-KAFOs and locked KAFOs) were compared to a wide variety of UC orthoses at baseline, not necessarily of the same type (Fig 2). Due to these inter-individual differences between baseline and follow-up conditions, we did not expect every subject to show a reduction in net energy cost, a consideration that might also explain the lack of effects at the group level.

Regarding type of specialist care orthosis, specialist care AFOs led to a significant reduction (9.5%) in net energy cost compared to UC orthoses. This difference is somewhat larger than the (non-significant) 7% reduction in energy cost reported for polio survivors wearing hinged

dorsiflexion-restricting AFOs compared only to shoes [26], and similar to a 9.2% reduction in energy cost for stiffness-optimized dorsal leaf AFOs compared to usual care AFOs in NMD cases [27]. Considering that the specialist care AFOs in our study were not optimized for stiffness, a recommended method aimed at improving walking energy cost [27], the reduction in energy cost found in our study (9.5%) is both substantial and clinically relevant. Implementation of stiffness optimization in our orthotic practice may have led to an even greater energy cost reduction in this population, although it should be noted that our comparison included a variety of usual care AFOs and orthopaedic shoes rather than just usual care AFOs. The reduction in energy cost was likely due to reduced excessive ankle dorsiflexion and coupled knee flexion, as well as increased ankle power, as previously suggested to underlie improvements in walking energy cost associated with AFOs in NMD [26, 27]. We recommend that future studies explore the role biomechanical gait parameters play in relation to walking energy cost reduction in guideline-based specialised orthotic care.

With respect to perceived walking ability, specialist care AFO-related improvements in stability, fear of falling and overall satisfaction while walking are in agreement with a previous study evaluating specialist care AFOs for polio survivors with calf muscle weakness [26, 48]. Considering that our study compared specialist care AFOs to a variety of UC orthoses as provided in clinical practice, rather than walking with shoes as in previous studies, the improvements found can be considered meaningful. For specialist care KAFOs, the effects on perceived stability and satisfaction are in line with earlier improvements found for specialist care carbon composite KAFOs [23, 49]. When compared to objectively measured walking ability in a gait laboratory, perceived effects such as safety while walking on irregular surfaces may be more representative of walking in daily life and therefore of greater importance to patients. Overall, subjects felt safer, more stable, less exhausted and more satisfied while walking, and had a lower fear of falling when using a specialist care orthosis compared to a UC orthosis, corresponding to the clinically relevant attainment of personal goals as measured by GAS. Furthermore, these improvements may have contributed to the shift in functional ambulation level, which increased with a specialist care orthosis from 8% to 19% full community walkers, although the increased use of walking aids at follow-up should be considered in this shift. Mean satisfaction scores with the specialist care orthosis were comparable with previous studies using the D-QUEST [26, 49, 50].

A major strength of our study is that it shows the benefits of specialist care orthoses as provided in actual clinical practice to the largest cohort of patients with NMD so far. Nevertheless, several limitations need to be considered. First, the collection of data from actual clinical practice has led to missing data of subjects across outcomes. Second, follow-up measurements were performed after 3 months of specialist care orthosis use, whereas some UC orthoses had been in use for 2 years or more. Consequently, the effectiveness of UC orthoses at baseline might have declined due to orthosis wear and/or NMD progression. Finally, occasional later adjustments to the specialist care orthoses to improve functioning were based on the follow-up measurement, perhaps suggesting that ultimate effectivity might have been underestimated. In future research, treatment outcomes should be compared to a control group receiving usual orthotic care, preferably with randomization of interventions, allowing for a more accurate determination of the benefit of specialist orthotic care over usual care.

## Conclusion

In conclusion, our study showed that the provision of custom-made specialist care lower limb orthoses improves personal goals and perceived walking ability outcomes compared to UC orthoses. While no improvements were found in walking speed and net energy cost in

combined orthosis groups, specialist care AFOs reduce net energy costs by almost 10% on average. A randomized controlled study is now warranted to confirm these findings.

## Supporting information

**S1 Fig. Schematic presentation of the process description medical devices [34].**
(TIF)

**S1 File. SPSS dataset file containing all outcome data that support the reported results.**
(SAV)

**S1 Checklist. STROBE statement—checklist of items that should be included in reports of observational studies.**
(DOCX)

## Author Contributions

**Conceptualization:** Elza van Duijnhoven, Fieke S. Koopman, Frans Nollet, Merel-Anne Brehm.

**Data curation:** Hilde E. Ploeger.

**Formal analysis:** Elza van Duijnhoven.

**Investigation:** Elza van Duijnhoven, Hilde E. Ploeger, Frans Nollet, Merel-Anne Brehm.

**Methodology:** Elza van Duijnhoven, Fieke S. Koopman, Hilde E. Ploeger, Frans Nollet, Merel-Anne Brehm.

**Project administration:** Elza van Duijnhoven, Merel-Anne Brehm.

**Supervision:** Fieke S. Koopman, Frans Nollet, Merel-Anne Brehm.

**Writing – original draft:** Elza van Duijnhoven.

**Writing – review & editing:** Elza van Duijnhoven, Fieke S. Koopman, Hilde E. Ploeger, Frans Nollet, Merel-Anne Brehm.

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
