## [Decision Letter · Decision Letter 0]

23 Sep 2022

PONE-D-22-15251Effects of specialist care leg orthoses on personal goal attainment and walking ability in adults with neuromuscular disordersPLOS ONE

Dear Dr. van Duijnhoven,

Thank you for submitting your manuscript to PLOS ONE. After careful consideration, we feel that it has merit but does not fully meet PLOS ONE’s publication criteria as it currently stands. Therefore, we invite you to submit a revised version of the manuscript that addresses the points raised during the review process.

Three reviewers have now evaluated your submission. They are positive about the manuscript, but have each identified some opportunities to strengthen it through a revision. Please refer to their comments below for more detail, and ensure that you respond to all of their suggestions when preparing your revision.

We look forward to receiving your revised manuscript.

Kind regards,

Jamie Males

Editorial Office

PLOS ONE

Journal Requirements:

2. Please include your full ethics statement in the ‘Methods’ section of your manuscript file. In your statement, please include the full name of the IRB or ethics committee who approved or waived your study, as well as whether or not you obtained informed written or verbal consent. If consent was waived for your study, please include this information in your statement as well

Reviewers' comments:

Reviewer's Responses to Questions

**Comments to the Author**

1. Is the manuscript technically sound, and do the data support the conclusions?

Reviewer #1: Yes

Reviewer #2: No

Reviewer #3: Yes

2. Has the statistical analysis been performed appropriately and rigorously? 

Reviewer #1: Yes

Reviewer #2: Yes

Reviewer #3: Yes

3. Have the authors made all data underlying the findings in their manuscript fully available?

Reviewer #1: Yes

Reviewer #2: Yes

Reviewer #3: Yes

4. Is the manuscript presented in an intelligible fashion and written in standard English?

Reviewer #1: Yes

Reviewer #2: Yes

Reviewer #3: Yes

5. Review Comments to the Author

Reviewer #1: This study compares usual orthotic care with a specific orthotic treatment approach in adults with neuromuscular conditions. The study finds no significant differences at a group level, but finds a significant subgroup difference (“specialist care AFOs reduce net energy costs by almost 10% on average”).

The study appears well conducted and the manuscript is well written; I thank the authors for creating this useful contribution to the literature. Strengths include the sample size (n=140), use of prospectively collected data, multiple outcome measures and the use of Strengthening the Reporting of Observational Studies in Epidemiology (STROBE) guidelines. The evaluation of a specific approach to orthotic prescription is useful, and develops previous work by the group.

While the paper is generally strong, I have provided comments below which may be useful if the article is revised:

Line 26 and throughout: “leg orthosis” – this is a stylistic comment, and may be due to preference, but ‘leg orthosis’ does not read well to a native English speaker and I suggest that ‘lower limb orthosis’ would be a better generic term.

Line 56: “In NMD, leg orthoses for lower extremity muscle weakness are intended to…” – this section introduces the use of lower limb orthoses. In order to set the context and justify the study, it may be worth considering references which demonstrate that orthoses are commonly used in adults with neurological conditions. See below for examples.

Young J, Moss C. Orthotic care needs in a cohort of neurological rehabilitation inpatients. Disabil Rehabil Assist Technol [Internet]. 2019 Nov 20;1–5. Available from: https://www.tandfonline.com/doi/full/10.1080/17483107.2019.1685018

Hada T, Momosaki R, Abo M. Impact of orthotic therapy for improving activities of daily living in individuals with spinal cord injury: a retrospective cohort study. Spinal Cord. 2018; 56(8):790–795.

Line 60-61: The acronyms ‘AFO’ and ‘KAFO’ are used. Suggest citing the ISO source, here is the weblink to the most recent version:

https://www.iso.org/obp/ui/#iso:std:iso:8549:-3:ed-2:v1:en

Line 72-73: “…a Dutch guideline for the prescription of leg orthoses in NMD was published in 2012” – given that the main premise of the study is to evaluate the special approach referenced here, and that this approach has only been published in Dutch, is it possible to give more detail on the process, or to reproduce aspects of the protocols as an appendix?

End of review.

Reviewer #2: 1- The purpose of this study was to compare the specialist care leg orthoses to usual care orthoses in terms of personal goal attainment, walking ability outcomes, and satisfaction. The main weakness of this study is the research method and presentation of data.

2- If you want to compare usual orthotic care with the orthoses made based on specific guidance, the orthoses type should be constant. As a result, this study method does not answer your research question.

3- Line 72- It is recommended to provide a brief text about the guidance provided in Dutch.

4- Lines 96-8, and now do you have an ethical code for the presented research?

5- Lines 111-134, you should provide definitions of your interventions precisely, in such a way that another researcher could repeat your investigation according to these definitions.

6- Lines 136-7, same as the previous comment.

7- Line 125, how did you perform the “3D gait analysis”? What instrument? Analysis process?

8- Line 144, Please provide a reference for the muscle strength test, its validity, and reliability.

9- Line 212, The information provided in this table is ambiguous. Please dived it into two separate tables.

10- Line 302, colored table!! Please change it to black and white.

11- Lines 397-8, your results can not be interpreted in this statement, because of comment 2.

Reviewer #3: One suggestion for improvement. Include the scope in which 'effectiveness' is evaluated in the Abstract  To examine the effectiveness "on ...?" of specialist care leg orthoses compared to usual care orthoses.

6. PLOS authors have the option to publish the peer review history of their article (what does this mean?). If published, this will include your full peer review and any attached files.

Reviewer #1: No

Reviewer #2: **Yes: **Kourosh Barati

Reviewer #3: No

---

## [Author Response · Author response to Decision Letter 0]

20 Oct 2022

Reviewers comments to author

Reviewer 1

The study appears well conducted and the manuscript is well written; I thank the authors for creating this useful contribution to the literature. Strengths include the sample size (n=140), use of prospectively collected data, multiple outcome measures and the use of Strengthening the Reporting of Observational Studies in Epidemiology (STROBE) guidelines. The evaluation of a specific approach to orthotic prescription is useful, and develops previous work by the group. While the paper is generally strong, I have provided comments below which may be useful if the article is revised:

Line 26 and throughout: “leg orthosis” – this is a stylistic comment, and may be due to preference, but ‘leg orthosis’ does not read well to a native English speaker and I suggest that ‘lower limb orthosis’ would be a better generic term.

Response: We changed leg orthosis into lower limb orthosis throughout the manuscript accordingly. 

Line 56: “In NMD, leg orthoses for lower extremity muscle weakness are intended to…” – this section introduces the use of lower limb orthoses. In order to set the context and justify the study, it may be worth considering references which demonstrate that orthoses are commonly used in adults with neurological conditions. See below for examples.

Young J, Moss C. Orthotic care needs in a cohort of neurological rehabilitation inpatients. Disabil Rehabil Assist Technol [Internet]. 2019 Nov 20;1–5. Available from: https://www.tandfonline.com/doi/full/10.1080/17483107.2019.1685018

Hada T, Momosaki R, Abo M. Impact of orthotic therapy for improving activities of daily living in individuals with spinal cord injury: a retrospective cohort study. Spinal Cord. 2018; 56(8):790–795.

Response: We acknowledge the suggestion to consider references which demonstrate that orthoses are commonly used in neuromuscular disorders. However, since our study specifically focusses on neuromuscular disorders and not on neurological disorders, we provided the study of O’Connor et al., 2016 as a reference instead (reference 11). 

Line 60-61: The acronyms ‘AFO’ and ‘KAFO’ are used. Suggest citing the ISO source, here is the weblink to the most recent version:

https://www.iso.org/obp/ui/#iso:std:iso:8549:-3:ed-2:v1:en

Response: As suggested, we have added the ISO source as a reference to our manuscript (reference 20).

Line 72-73: “…a Dutch guideline for the prescription of leg orthoses in NMD was published in 2012” – given that the main premise of the study is to evaluate the special approach referenced here, and that this approach has only been published in Dutch, is it possible to give more detail on the process, or to reproduce aspects of the protocols as an appendix?

Response: The intervention indeed concerns the provision of leg orthoses according to a Dutch guideline. For clarification to the reader, we have added some details to the description of the guideline in the introduction (page 5, lines 75-79), and we have extended the description of the provision process in the methods section (page 6-7, lines 116-120).

Reviewer 2 

1- The purpose of this study was to compare the specialist care leg orthoses to usual care orthoses in terms of personal goal attainment, walking ability outcomes, and satisfaction. The main weakness of this study is the research method and presentation of data.

2- If you want to compare usual orthotic care with the orthoses made based on specific guidance, the orthoses type should be constant. As a result, this study method does not answer your research question.

Response: We assessed the effects of following a specific guideline that was developed to standardize the provision of leg orthoses in neuromuscular disorders, not limited to a specific type of orthosis. Therefore, we choose to include all patients who were provided with leg orthoses that are described in the guideline, regardless of type, to be able to evaluate its effects compared to usual care. We agree that a comparison with a constant orthosis type would be ideal as working mechanisms differ. However, the majority of subjects used another type of leg orthosis at baseline (usual care orthosis) than at follow-up (specialist care orthosis), because, according to the guideline-based approach, a different type was indicated (Table 1). Therefore, we feel confident to have addressed our research question. Furthermore, in order to provide some insights into the effects of different subtypes of leg orthoses, we have included separate analyses for subjects that were provided with specialized AFOs, KAFOs, and SC-KAFOs respectively. 

3- Line 72- It is recommended to provide a brief text about the guidance provided in Dutch.

Response: We acknowledge the recommendation and provided some additional information in the introduction (page 5, lines 75-79) and to the paragraph describing the guideline (page 6-7, lines 116-120).

4- Lines 96-8, and now do you have an ethical code for the presented research?

Response: The requirement for ethical review for the presented research under the Medical Research Involving Human Subjects Act in the Netherlands was waived by our local ethics committee (under study reference: W21_437), because participants were not subjected to procedures or were required to follow rules of behavior.

5- Lines 111-134, you should provide definitions of your interventions precisely, in such a way that another researcher could repeat your investigation according to these definitions.

Response: The intervention concerns specialized orthotic care, i.e. providing leg orthoses according to the Dutch guideline. In the methods section (page 6-7, lines 116-144), we have explained the protocol that patients complete during the provision process as extensive as possible to allow replication. Furthermore, the reference for the full guideline is given in the text. 

6- Lines 136-7, same as the previous comment.

Response: For this study on specialist care orthoses, we analyzed data that were previously collected during orthotic care delivery in our rehabilitation outpatient clinic. For a comparison with usual orthotic care, we investigated differences in effects between the specialist care orthoses and the usual care orthoses (i.e. the orthoses that subjects used to wear at baseline). These usual care orthoses were all provided at rehabilitation centers or hospitals elsewhere (who did not implement the guideline), and information about the local provision processes was not available. 

7- Line 125, how did you perform the “3D gait analysis”? What instrument? Analysis process?

Response: The 3D gait analysis was performed using a VICON motion capture system with 12 cameras that captured subjects’ gait through markers according to the Plug In Gait model, and two force plates. For clarification, we have added this information to our manuscript (page 7, lines 128-136). 

8- Line 144, Please provide a reference for the muscle strength test, its validity, and reliability.

Response: As suggested, we added a reference for the methodology of manual muscle testing (reference 37), and for its validity and reliability (reference 38).

9- Line 212, The information provided in this table is ambiguous. Please dived it into two separate tables.

Response: We moved the information on frequency of wearing the specialist care orthosis to Table 2, to avoid ambiguity in Table 1.

10- Line 302, colored table!! Please change it to black and white.

Response: We changed the colors in Table 4 to black and white tones accordingly. 

11- Lines 397-8, your results can not be interpreted in this statement, because of comment 2.

Response: We concluded that the provision of custom made specialist care orthoses improves personal goals and perceived walking ability outcomes compared to UC orthoses. Although it would be interesting to provide a more detailed overview of the effects of guidance-based versus usual orthotic care in patients that receive the same type of orthosis, we feel confident that we can draw this conclusion from results in a heterogenic population of people who were provided with different types of leg orthoses, but all according to the same guideline and corresponding treatment process.

Reviewer 3 

One suggestion for improvement. Include the scope in which 'effectiveness' is evaluated in the Abstract  To examine the effectiveness "on ...?" of specialist care leg orthoses compared to usual care orthoses.

Response: As suggested, we have added on which outcomes the effectiveness was evaluated.

---

## [Decision Letter · Decision Letter 1]

5 Dec 2022

Effects of specialist care lower limb orthoses on personal goal attainment and walking ability in adults with neuromuscular disorders

PONE-D-22-15251R1

Dear Dr. van Duijnhoven,

We’re pleased to inform you that your manuscript has been judged scientifically suitable for publication and will be formally accepted for publication once it meets all outstanding technical requirements.

Kind regards,

George Vousden

Staff Editor

PLOS ONE

Additional Editor Comments (optional):

Reviewers' comments:

Reviewer's Responses to Questions

**Comments to the Author**

1. If the authors have adequately addressed your comments raised in a previous round of review and you feel that this manuscript is now acceptable for publication, you may indicate that here to bypass the “Comments to the Author” section, enter your conflict of interest statement in the “Confidential to Editor” section, and submit your "Accept" recommendation.

Reviewer #2: All comments have been addressed

2. Is the manuscript technically sound, and do the data support the conclusions?

Reviewer #2: Partly

3. Has the statistical analysis been performed appropriately and rigorously? 

Reviewer #2: Yes

4. Have the authors made all data underlying the findings in their manuscript fully available?

Reviewer #2: Yes

5. Is the manuscript presented in an intelligible fashion and written in standard English?

Reviewer #2: Yes

6. Review Comments to the Author

Reviewer #2: (No Response)

7. PLOS authors have the option to publish the peer review history of their article (what does this mean?). If published, this will include your full peer review and any attached files.

Reviewer #2: No

---

## [Editor Report · Acceptance letter]

27 Dec 2022

PONE-D-22-15251R1 

Effects of specialist care lower limb orthoses on personal goal attainment and walking ability in adults with neuromuscular disorders 

Dear Dr. van Duijnhoven:

I'm pleased to inform you that your manuscript has been deemed suitable for publication in PLOS ONE. Congratulations! Your manuscript is now with our production department. 

Kind regards, 

on behalf of

Dr. George Vousden 

Staff Editor

PLOS ONE